# Design Method for a Wideband Non-Uniformly Spaced Linear Array Using the Modified Reinforcement Learning Algorithm

**DOI:** 10.3390/s22145456

**Published:** 2022-07-21

**Authors:** Seyoung Kang, Seonkyo Kim, Cheolsun Park, Wonzoo Chung

**Affiliations:** 1Department of Computer Science and Engineering, Korea University, Seoul 02841, Korea; sykang0229@korea.ac.kr; 2Agency for Defense Development, Daejeon 34186, Korea; skkim0416@add.re.kr (S.K.); csun1402@add.re.kr (C.P.); 3Department of Artificial Intelligence, Korea University, Seoul 02841, Korea

**Keywords:** reinforcement learning (RL), non-uniformly spaced linear array (NUSLA), optimization

## Abstract

In this paper, we present a design method for a wideband non-uniformly spaced linear array (NUSLA), with both symmetric and asymmetric geometries, using the modified reinforcement learning algorithm (MORELA). We designed a cost function that provided freedom to the beam pattern by setting limits only on the beam width (BW) and side-lobe level (SLL) in order to satisfy the desired BW and SLL in the wide band. We added the scan angle condition to the cost function to design the scanned beam pattern, as the ability to scan a beam in the desired direction is important in various applications. In order to prevent possible pointing angle errors for asymmetric NUSLA, we employed a penalty function to ensure the peak at the desired direction. Modified reinforcement learning algorithm (MORELA), which is a reinforcement learning-based algorithm used to determine a global optimum of the cost function, is applied to optimize the spacing and weights of the NUSLA by minimizing the proposed cost function. The performance of the proposed scheme was verified by comparing it with that of existing heuristic optimization algorithms via computer simulations.

## 1. Introduction

An array antenna enhances the sensitivity of signals received from specific directions by increasing the directivity and antenna efficiency through attenuation of interference from other directions [1,2,3,4]. Array antennas have been widely implemented in various applications such as wireless communication and radar owing to their advantages, such as high directivity, narrow beam width (BW), and low side-lobe level (SLL) [5,6,7,8,9]. Extensive research has been conducted on array antennas for narrowband signals, wherein they deal with only a single operating frequency. A uniform linear array (ULA) structure, wherein the antennas are spaced by the half-wavelength of the operating frequency, is typically used in narrowband arrays owing to its analytically tractable weight optimization.

Wideband array antennas, which handle several frequencies over a wide range, have gained considerable attention in recent years for military and commercial applications [10,11,12]. Various approaches have been implemented to optimize the weights of a fixed-spacing ULA over a wide range of frequencies [13,14,15,16]. These approaches include the least squares approach for the optimal Riblet–Chebyshev weights [13], a fast Fourier transform-based algorithm for frequency-invariant beamforming (FIB) [14], a window-based method that enforces a constant beam width [15], and a singular value decomposition approach for FIB [16].

However, the performance of these approaches is limited by the fundamental limits of the ULA; that is, its uniform spacing is optimized for a narrow band. Therefore, a non-uniformly spaced linear array (NUSLA) has been developed as an alternative. The NUSLA was initially developed to overcome the trade-off between the BW and SLL of the narrowband ULA. Several heuristic algorithms, such as simulated annealing (SA) [17], genetic algorithm (GA) [18], firefly algorithm (FA) [19], particle swarm optimization (PSO) [20], and salp swarm algorithm (SSA) [21], have been proposed to optimize the weights and spacing of the NUSLA in an analytic form.

The NUSLA presents considerable potential for wideband applications; however, only a few studies have been conducted to optimize both the spacing and weights of the NUSLA over a wide band. Several optimization schemes have been developed based on analytic approximations, such as the generalized matrix pencil method [22] and unitary matrix pencil method [23]. However, these approximation approaches are suboptimal owing to the complexity of the structure of NUSLA.

Although there have been several heuristic optimization studies applied for antenna array, the works are focused on narrowband NUSLA. For example, null generation in narrowband NUSLA by using various heuristic optimization algorithms such as bat algorithm (BA) [24], firework algorithm (FWA) [25], and modified ant lion optimization (MALO) [26].

The spacing and weights of a wideband NUSLA can be optimized by employing various heuristic optimization algorithms, such as firefly algorithm (FA) [19], salp swarm algorithm (SSA)[21], and quantum particle swarm optimization (QPSO) [27]. However, these heuristic optimization algorithms face a significant drawback in that convergence to the global optimum cannot be ensured, despite the trade-off between the convergence time and the chance to achieve the global optimum. Parameter optimization was performed for the QPSO in [27] via trial-and-error to prevent falling into a local optimum. However, the performance of QPSO is considerably affected by the hyper-parameter setting, which is typically optimized by trial-and-error on a case-by-case basis (such as scanning angles and element size). Therefore, a new approach must be developed for the optimization of the wideband NUSLA, which can provide a more reliable and robust convergence property.

In this paper, we propose a novel algorithm based on reinforcement learning (RL) to optimize the spacing and weights of a wideband NUSLA. Several algorithms based on RL have been developed in recent years and have achieved remarkable performance improvements in various fields. To the best of our knowledge, this is the first wideband NUSLA optimization approach using RL. A global minimum finding algorithm based on RL, known as the modified reinforcement learning algorithm (MORELA) [28], presents a significant advantage over existing heuristic algorithms. The algorithm is less insensitive to the hyper-parameter setting, demonstrates a higher probability of finding the global optimum, and is more efficient for high-dimensional cost functions.

Conventional cost functions reduce the difference between a desired pencil beam pattern and the array beam pattern. In contrast, we propose a novel cost function that increases the degree of freedom for the formation of a desired beam pattern by penalizing only the SLL and not specifying the beam pattern. This approach increases the degree of freedom of the weights and contributes to the improvement in the overall performance. With this novel cost function and the optimization approach based on RL, the proposed algorithm achieves the state-of-the-art performance with a simple hyper-parameter setting and relatively efficient computational complexity.

The remainder of this paper is organized as follows. Section 2 describes the array model, and Section 3 introduces the proposed cost functions. Section 4 presents the modified RL algorithm employed to minimize the cost function. In Section 5, we present an extensive analysis of the performance of the proposed algorithm using numerical simulations. Section 6 presents the conclusions of the study.

## 2. Antenna Array Model

Consider a NUSLA with *N* isotropic elements located on y-axis, as presented in Figure 1.

Let $d_{n}$ denote the spacing between the origin and *n*-th element, and let $w_{n}\left(=a_{n}e^{j\phi_{n}}\right)$ denote the complex weight of the *n*-th element with amplitude, $a_{n}$, and phase, $\phi_{n}$.

Array factor (AF) scanning along the $\theta_{0}$ direction at an operating frequency, *f*, denotes the beam pattern of the array and is described as follows [2]:(1)$$\begin{array}{c} AF\left(\theta ;w,d,f,\theta_{0}\right)=\sum_{n=1}^{N}w_{n}\exp\left(jkd_{n}\left(\sin\left(\theta \right)-\sin\left(\theta_{0}\right)\right)\right), \end{array}$$
where θ denotes the polar angle from the *z*-axis (−π/2∼π/2), k(=2π/λ) denotes the wavenumber, λ(=c/f) denotes wavelength, and *c* denotes the propagation speed. Conventionally, the beamformer adjusted all amplitudes and phases of each antenna to synthesize the beams, as presented in Figure 2.

Once a signal arrives at the antenna array consisted of *N* elements ($\{A_{n}\}$), the beamformer multiplies the output signal of each element, $x_{n}$, with amplitudes ($a_{n}$) and phases ($\phi_{n}$), and then synthesizes them ($y=\sum_{n=1}^{N}x_{n}a_{n}e^{j\phi_{n}}$). The complex number $w_{n}:=a_{n}e^{j\phi_{n}}$ is called the weight of the *n*-th element.

If the geometric configuration of NUSLA is symmetric with respect to the origin, the spacing and weights of the NUSLA are symmetrically allocated on both sides of the origin, and all weights emerge as real values, $w_{n}=a_{n}$. Furthermore, when *N* is an odd number, the center element must be located at the origin to maintain symmetry. Therefore, the AF of a symmetric NUSLA can be rewritten as:(2)$$\begin{array}{c} AF\left(\theta ;w,d,f,\theta_{0}\right)=\left\{\begin{array}{cc} a_{0}+2\sum_{n=1}^{M}a_{n}\cos\left(kd_{n}\left(\sin\left(\theta \right)-\sin\left(\theta_{0}\right)\right)\right) & \mathrm{for}\;N=2M+1\\ 2\sum_{n=1}^{M}a_{n}\cos\left(kd_{n}\left(\sin\left(\theta \right)-\sin\left(\theta_{0}\right)\right)\right) & \mathrm{for}\;N=2M, \end{array}\right. \end{array}$$
where *M* denotes the number of elements on one side with respect to the origin.

Spacing and weight optimization is performed to determine the optimal spacing vector, $d=[d_{1},\cdots ,d_{N}]$, and weight vector, $w=[w_{1},\cdots ,w_{N}]$, such that the resultant beam described by the AF, AF(θ), satisfies the given BW and SLL constraint.

## 3. Cost Function

In this section, we propose a cost function to obtain the desired beam pattern for the NUSLA. The cost function for a narrow-band beam pattern with a given BW and SLL for a given scanning angle was first introduced and extended for wide-band operation. An additional penalty term was introduced for the asymmetric configuration of the array to ensure that an exact scanning angle was obtained in the desired direction.

### 3.1. Cost Function for Narrowband NUSLAs

Given a BW of the desired beam (in radian), SLL (in dB), and scanning angle $\theta_{0}$, the cost function can be written as:(3)$$\begin{array}{c} f_{cost}\left(w,d;f,\theta_{0}\right)=\int_{I}\left|SLL-20\log_{10}\left(\frac{\left|AF(\theta ,w,d;f,\theta_{0})\right|}{\max\left(\left|AF(\theta ,w,d;f,\theta_{0})\right|\right)}\right)\right|\;d\theta , \end{array}$$
where the integration interval, I, is given as:(4)$$\begin{array}{c} I=\left[-\frac{\pi }{2},\theta_{0}-\frac{BW}{2}\right]⋃\left[\theta_{0}+\frac{BW}{2},\frac{\pi }{2}\right]. \end{array}$$

This cost function only affects the SLL outside the desired main BW and does not specify the main beam pattern, which may provide more freedom to yield low SLL in comparison to the conventional cost function, which enforces the main beam pattern shape. Since AF is a superposition of complex sinusoidal with different phases, the resulting beam exhibits a pencil shape, particularly for a symmetric NUSLA. An additional penalty term is added for an asymmetric NUSLA to ensure that the peak of the AF is located at the desired position:(5)$$\begin{array}{c} f_{assym}\left(w,d;f,\theta_{0}\right)=f_{cost}\left(w,d;f,\theta_{0}\right)+P\left(w,d;f,\theta_{0}\right) \end{array}$$
where
(6)$$\begin{array}{c} P\left(w,d;f,\theta_{0}\right)=\left\{\begin{array}{cc} \varepsilon , & \mathrm{for}\arg\max_{\theta }AF\left(\theta ,w,d;f,\theta_{0}\right)\neq \theta_{0}\\ 0, & \mathrm{for}\;\mathrm{otherwise}. \end{array}\right. \end{array}$$

Figure 3 illustrates the advantages of the proposed cost function, where the red areas represent the cost values, i.e., the difference between the target pattern and the generated pattern, of the cost functions. Figure 3a illustrates an example of a conventional cost function (used in [27]), which consists of a main pencil beam and a flat side-lobe. Hence, the cost is affected by the shape of the design main pencil beam. Figure 3b presents the proposed cost function, which penalizes only the SLL that is larger than the given SLL and, therefore, the freedom to build main beam is expected to yield performance gain in lowering the SLL.

### 3.2. Cost Function for Wideband NUSLAs

Beam distortion occurs whenever a narrowband array antenna operates outside the designed frequency. The spacing and weights of the NUSLA must be optimized such that the desired BW and SLL are satisfied for the design frequency band, $\left[f_{min},f_{max}\right]$, to minimize beam distortion. The cost function for a wideband NUSLA is defined as an extension of the narrowband NUSLA cost function, which can be written as follows:(7)$$\begin{array}{c} f_{wide}\left(w,d;\theta_{0}\right)=\sum_{i=1}^{N_{f}}c_{i}f_{cost}\left(w,d;f_{min}+\frac{f_{max}-f_{min}}{N_{f}-1}\left(i-1\right),\theta_{0}\right), \end{array}$$
where $N_{f}\left(\geq 2\right)$ denotes the number of frequencies considered in the given frequency range $\left[f_{min},f_{max}\right]$, and $c_{i}\left(\geq 0\right)$ denotes the weight of the *i*-th frequency.

For the asymmetric NUSLA, the penalty term defined in (Equation 6) is considered as follows:(8)$$\begin{array}{c} f_{wide,assym}\left(w,d;\theta_{0}\right)=\sum_{i=1}^{N_{f}}c_{i}f_{cost}\left(w,d;f_{min}+\frac{f_{max}-f_{min}}{N_{f}-1}\left(i-1\right),\theta_{0}\right)+P\left(w,d;f,\theta_{0}\right). \end{array}$$

The weight, $c_{i}$, reflects the importance of a certain frequency, and in this study, we set $c_{i}=1$ for all $i=1,\cdots ,N_{f}$.

## 4. Optimization Algorithm Based on Reinforcement Learning

This section discusses an optimization scheme for the proposed algorithm based on the RL approach. The RL approach exhibits a higher probability of finding a global optimum than existing heuristic optimization algorithms owing to its search and reward characteristics. The MORELA [28] is a global optimum-finding algorithm, which is based on the model-free *Q*-learning-based RL approach. One advantage of the MORELA is the use of a sub-environment that is generated around the best solution determined in the previous learning step, and it plays an important role in the prevention of falling into a local optima by searching around the best solution. The MORELA comprises several parameters (T,K,α,γ,β). Here, *T* represents the number of maximum learning episodes, *K* represents the size of the sub-environment, α represents the learning rate, γ represents the discounting (or forgetting) factor, and β represents the search space-reducing factor. The sub-environment explores a specific interval around the best solutions in the previous learning episode, and the interval decreases in proportion to search space-reducing factor as the learning episode progresses.

Firstly, at the 0-th learning episode, *K* different initial guesses of the spacing parameter vectors ($d_{k}^{\left(0\right)}$) and complex weight parameter vectors $w_{k}^{\left(0\right)}$ that are given by amplitude vectors ($a_{k}^{\left(0\right)}$) and phase vectors $\left(\phi_{k}^{\left(0\right)}\right)$ are randomly generated as follows:(9)$$\begin{array}{c} \begin{array}{cc} d_{k,n}^{\left(0\right)} & =\sum_{i=1}^{n}\left(d_{min}+\left(d_{max}-d_{min}\right)\times u_{k,i}^{\left(0\right)}\right)\\ a_{k,n}^{\left(0\right)} & =a_{min}+\left(a_{max}-a_{min}\right)\times v_{k,n}^{\left(0\right)}\\ \phi_{k,n}^{\left(0\right)} & =\phi_{min}+\left(\phi_{max}-\phi_{min}\right)\times s_{k,n}^{\left(0\right)} \end{array}\;\;k=1,\cdots ,K, \end{array}$$
where n(=1,⋯,N) denotes the order number of array elements, $d_{min}$ and $d_{max}$ denote the minimum and maximum spacing between the neighbor array elements, respectively, $a_{min}$ and $a_{max}$ denote the minimum and maximum amplitude of weights, respectively, $\phi_{min}$ and $\phi_{max}$ denote the minimum and maximum phase of weights, respectively, and $u_{k}^{\left(0\right)}={\left[u_{k,1}^{0},\cdots ,u_{k,N}^{\left(0\right)}\right]}^{T}$, $v_{k}^{\left(0\right)}={\left[v_{k,1}^{0},\cdots ,v_{k,N}^{\left(0\right)}\right]}^{T}$, and $s_{k}^{\left(0\right)}={\left[s_{k,1}^{0},\cdots ,s_{k,N}^{\left(0\right)}\right]}^{T}$ denote the vectors of independent standard uniform random numbers between 0 and 1 at the 0-th learning episode. Let $\left(d_{best}^{\left(0\right)},w_{best}^{\left(0\right)}=a_{best}^{\left(0\right)}⊙e^{j\phi_{best}^{\left(0\right)}}\right)$ denote the best performing parameter vector among *K* initial vectors in terms of minimizing the cost function, where ⊙ denotes the element-wise multiplication. The set of K+1 vectors: (10)$$\begin{array}{c} \left\{\left(d_{k,n}^{\left(0\right)},w_{k,n}^{\left(0\right)}=a_{k,n}^{\left(0\right)}e^{j\phi_{k,n}^{\left(0\right)}}\right),\left(d_{best,n}^{\left(0\right)},w_{best,n}^{\left(0\right)}\right)|k=1,\cdots ,K,n=1,\cdots ,N,\right\} \end{array}$$
comprises the *Q*-table at the 0-th episode.

Let us consider $\left\{\left(d_{k,n}^{\left(t\right)},w_{k,n}^{\left(t\right)}=a_{k,n}^{\left(t\right)}e^{j\phi_{k,n}^{\left(t\right)}}\right),\left(d_{best,n}^{\left(t\right)},w_{best,n}^{\left(t\right)}\right)|k=1,\cdots ,K,n=1,\cdots ,N\right\}$ as the *Q*-table for the *t*-th episode. The original *Q*-learning-type RL generates the next *Q*-table solely based on the current *Q*-table. However, the MORELA generates a sub-environment by sampling *K* additional sets of parameter vectors near the best parameters. The *K* additional parameter vectors are given as follows:(11)$$\begin{array}{c} \begin{array}{cc} d_{K+k,n}^{\left(t\right)} & =\sum_{i=1}^{n}\left(d_{best,i}^{\left(t\right)}-d_{best,i-1}^{\left(t\right)}+\left(d_{max}-d_{min}\right)\left(2u_{K+k,i}^{\left(t\right)}-1\right)\beta^{t}\right)\\ a_{K+k,n}^{\left(t\right)} & =a_{best,n}^{\left(t\right)}+\left(a_{max}-a_{min}\right)\left(2v_{K+k,n}^{\left(t\right)}-1\right)\beta^{t}\\ \phi_{K+k,n}^{\left(t\right)} & =\phi_{best,n}^{\left(t\right)}+\left(\phi_{max}-\phi_{min}\right)\left(2s_{K+k,n}^{\left(t\right)}-1\right)\beta^{t} \end{array},k=1,\cdots ,K. \end{array}$$

The sub-environment interval decreases with the increase in the number of learning episodes, owing to the $\beta^{t}$ term for 0<β<1. The cost function values at the 2K+1 parameter vectors, including the vectors in the sub-environment, are evaluated and the worst *K* parameter vectors are removed and the best performing parameter vector is saved as: $\left(d_{best}^{\left(t+1\right)},a_{best}^{\left(t+1\right)},\phi_{best}^{\left(t+1\right)}\right)$. Then, the survived *K* parameter vectors are updated as follows:(12)$$\begin{array}{c} \begin{array}{cc} d_{k}^{\left(t+1\right)} & =\left(1-\alpha \right)\times d_{k}^{\left(t\right)}+\alpha \left(r_{k,d}^{\left(t+1\right)}+\gamma \times d_{best}^{\left(t\right)}\right)\\ a_{k}^{\left(t+1\right)} & =\left(1-\alpha \right)\times a_{k}^{\left(t\right)}+\alpha \left(r_{k,a}^{\left(t+1\right)}+\gamma \times a_{best}^{\left(t\right)}\right)\\ \phi_{k}^{\left(t+1\right)} & =\left(1-\alpha \right)\times \phi_{k}^{\left(t\right)}+\alpha \left(r_{k,\phi }^{\left(t+1\right)}+\gamma \times \phi_{best}^{\left(t\right)}\right), \end{array} \end{array}$$
where the reward terms are computed as follows:(13)$$\begin{array}{c} r_{k,d}^{\left(t+1\right)}=\frac{d_{best}^{\left(t+1\right)}-d_{k}^{\left(t\right)}}{d_{k}^{\left(t\right)}},\;r_{k,a}^{\left(t+1\right)}=\frac{a_{best}^{\left(t+1\right)}-a_{k}^{\left(t\right)}}{a_{k}^{\left(t\right)}},\;r_{k,\phi }^{\left(t+1\right)}=\frac{\phi_{best}^{\left(t+1\right)}-\phi_{k}^{\left(t\right)}}{\phi_{k}^{\left(t\right)}}. \end{array}$$

This process defines the *Q*-table for the (t+1)-th learning episode. Algorithm 1 presents the detailed process of the proposed method to optimize the spacing and weights of a wideband NUSLA using the MORELA.
**Algorithm 1** An algorithm for optimizing NUSLA using MORELA.**Initialization Step**Initialize parameters (T,K,α,β,γ)Generate randomly *K* parameter vectors ($d_{k}^{\left(0\right)}$ and $w_{k}^{\left(0\right)}$) using Equation (Equation 9)**for**k=1,⋯,KCalculate cost function (Equation 8) for $d_{k}^{\left(0\right)}$ and $w_{k}^{\left(0\right)}$**end for**Store the best performing parameters ($d_{best}^{\left(0\right)}$ and $w_{best}^{\left(0\right)}$)**Update Step****for**t=1,⋯,TGenerate additional *K* parameter vectors ($d_{K+k}^{\left(t\right)}$ and $w_{K+k}^{\left(t\right)}$) using Equation (Equation 11)**for**k=1,⋯,KCalculate the cost function (Equation 8) for original ($d_{k}^{\left(t\right)}$ and $w_{k}^{\left(t\right)}$) and sub-environment ($d_{K+k}^{\left(t\right)}$ and $w_{K+k}^{\left(t\right)}$)**end for**Remove the worst *K* parameter vectorsStore the best performing parameters as ($d_{best}^{\left(t+1\right)}$ and $w_{best}^{\left(t+1\right)}$)Calculate the *K* reward vectors using Equation (Equation 13)Update the survived *K* parameter vectors using Equation (Equation 12)**end for**

## 5. Simulation Results

In this section, we discuss the performance of the proposed algorithm, which is determined through a numerical simulation and performance comparison with existing methods based on other heuristic optimization algorithms, such as FA [19], SSA [21], MALO [26] and QPSO [27].

We have performed all simulations in the same setting by using MATLAB R2020a on a computer equipped with an Intel(R) Core(TM) i5-8600 CPU at 3.10GHz and 16GB of RAM. All parameters are presented in Table 1 and Table 2. Table 1 lists the detailed parameter settings to optimize the wideband NUSLA, wherein $\lambda_{min}$ denotes wavelength at the highest frequency. Table 2 lists the MORELA parameter settings.

The parameters for the other algorithms in comparison (such as FA, SSA, QPSO, and MALO) are set as presented in their own references. All algorithms are test at the same system. For each algorithm, we have repeatedly performed 100 simulations and have selected the best performing result among 100 results. The beamforming performance of the algorithms are evaluated in term of half-power beam width (HPBW) at the lowest frequency (Max HPBW) and at the highest frequency (Min HPBW), peak SLL (PSLL), and run time.

### 5.1. Wideband Symmetric NUSLA

This sub-section discusses the performance of proposed algorithm for a wideband NUSLA with symmetric geometry, which is determined by evaluating the HPBW and PSLL. We consider an array antenna equipped with 20 elements (N=20) and scanning angles of 0°, 30° and 60°. However, we optimized only (M=10) spacing parameters and real-valued weights (all phase are zero) owing to the symmetric structure. The desired SLL is set to −20 dB and the desired BW is set as 13°, 20° and 30° for the scanning angles of 0°, 30° and 60°, respectively.

Figure A1, Figure A2 and Figure A3 (Appendix A) present a comparison of the beam patterns obtained from the proposed algorithm for various frequencies (0.5 GHz–1 GHz) and several scanning angles ($\theta_{0}=0{}^{\circ},30{}^{\circ},60{}^{\circ}$) with the beam patterns produced by existing heuristic algorithms.

Table 3 presents the performance of the proposed algorithm in comparison with existing algorithms in terms of the HPBW, PSLL, and run time. The proposed algorithm presents the best performance among the existing methods. Although the SSA presents the shortest run time owing to its simple two-step structure, its performance is unsatisfactory. The FA presents relatively good performance; however, it is limited by a slow convergence time. Since MALO is hybrid of ALO and PSO with a chaotic map, it shows a partly superior performance than QPSO, but inferior performance than the proposed algorithm with slower convergence time. QPSO presents an excellent trade-off between the performance and run time; however, it is significantly affected by the setting of a hyper-parameter, called contraction and expansion coefficient, which controls the convergence speed and performance of QPSO. In [27], the contraction and expansion coefficient is determined between [0.5,0.8] using several formulas determined by trial-and-error for each case. Thanks to the nature of RL, the proposed algorithm is free from any complicated hyper-parameter setting. In this simulation, we used the hyper-parameter settings for QPSO described in [27], where they are found for each scanning angle ($\theta_{0}$) and element size (*N*) via trial-and-error. The proposed algorithm based on MORELA achieves the best performance without using a dedicated parameter optimization process and works with a single parameter setting for all cases. Although the running time of MORELA takes about twice of QPSO in Table 3, it is comparison under the given hyper-parameter. Note that QPSO requires at least several tries of the hyper-parameter optimization procedure and the overall computation complexity is higher than MORELA.

For wide-band optimization of NUSLA, we used $N_{f}=11$, i.e., the weights were optimized only for $N_{f}$ discrete frequency points between 0.5 GHz and 1 GHz. Figure 4 and Figure 5 present the HPBW and PSLL, respectively, for all frequencies between 0.5 GHz and 1 GHz. These figures demonstrate that the weights generate desirable beam patterns for all frequencies between 0.5 GHz and 1 GHz. Figure 5c indicates that QPSO fails at the scanning angle, $\theta_{0}=60{}^{\circ}$, operating above approximately 0.78 GHz.

### 5.2. Wideband Asymmetric NUSLA

In this sub-section, we discuss the performance of the proposed algorithm for a wideband NUSLA with an asymmetric geometry, which is determined by evaluating the HPBW and PSLL. We consider an array antenna equipped with 20 elements (N=20) and scanning angles of 0°, 30° and 60°. Unlike symmetric geometry, we have completely optimized (N=20) spacing parameters and the weights, which are now complex values. The desired SLL is set to −20 dB and the desired BW is set as 13°, 20° and 30° for the scanning angles of 0°, 30° and 60°, respectively. A penalty function with a penalty coefficient, $\varepsilon =10^{10}$, is used to place the peak of the beam pattern at the desired location.

Figure A4, Figure A5 and Figure A6 (Appendix A) present a comparison of the beam patterns obtained from the proposed algorithm for various frequencies (0.5 GHz–1 GHz) and scanning angles ($\theta_{0}=0{}^{\circ},30{}^{\circ},60{}^{\circ}$) with the beam patterns produced by existing heuristic algorithms.

Table 4 summarizes the performance of the proposed algorithm in comparison with that of existing heuristic algorithms. The proposed algorithm exhibits the best performance among existing methods, similar to the symmetric case. A wideband NUSLA with an asymmetric geometry requires more time than a wideband NUSLA with a symmetric geometry to optimize the spacing and weights.

For wide-band optimization of NUSLA, we used $N_{f}=11$. Figure 6 and Figure 7 show the performance of NULSA for the frequency range between 0.5 GHz and 1 GHz. Notice that in Figure 5c the proposed algorithm shows the best performance, while other heuristic algorithms suffer from performance degradation in the high frequency range.

## 6. Conclusions

In this paper, we proposed a novel design method for a wideband NUSLA based on RL. We designed an enhanced cost function that improved the degree of freedom to optimize the spacing and weights of the NUSLA. This cost function was optimized using the MORELA. The proposed method outperformed existing methods in terms of HPBW and PSLL. Furthermore, the proposed method is free from delicate hyper-parameter settings and saves overall computation time, unlike other hyper-parameter sensitive algorithms such as QPSO where the hyper-parameter optimization based on time consuming trial-and-error is crucial and inevitable for the optimal performance.

## Figures and Tables

**Figure 1 sensors-22-05456-f001:**
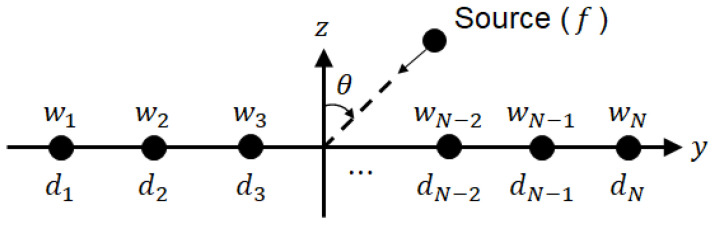
Geometry of asymmetric non-uniformly spaced linear array.

**Figure 2 sensors-22-05456-f002:**
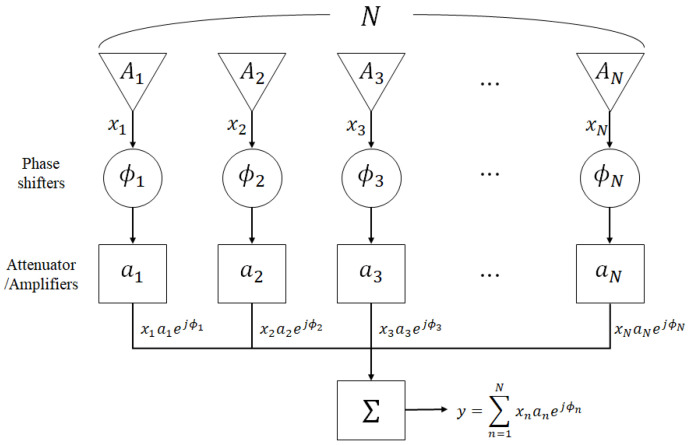
Schematic layout of array antenna.

**Figure 3 sensors-22-05456-f003:**
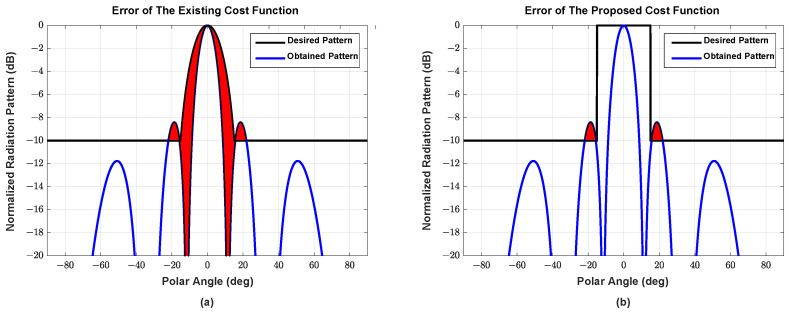
Two examples of (**a**) the existing cost function and (**b**) the proposed cost function.

**Figure 4 sensors-22-05456-f004:**
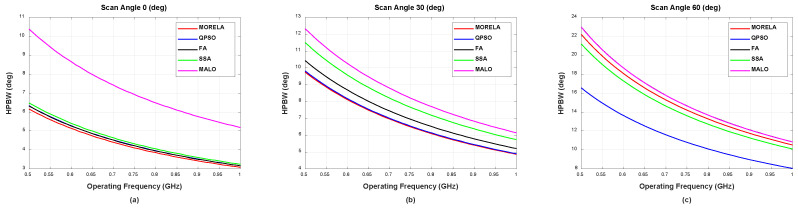
The HPBW performance of symmetric NUSLA in case for (**a**) $\theta_{0}=0{}^{\circ}$, (**b**) $\theta_{0}=30{}^{\circ}$, and (**c**) $\theta_{0}=60{}^{\circ}$.

**Figure 5 sensors-22-05456-f005:**
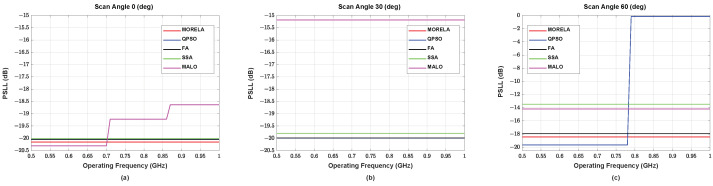
The SLL performance of symmetric NUSLA in case for (**a**) $\theta_{0}=0{}^{\circ}$, (**b**) $\theta_{0}=30{}^{\circ}$, and (**c**) $\theta_{0}=60{}^{\circ}$.

**Figure 6 sensors-22-05456-f006:**
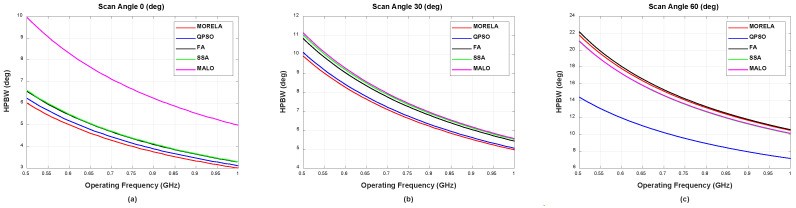
The HPBW performance of symmetric NUSLA in the case for (**a**) $\theta_{0}=0{}^{\circ}$, (**b**) $\theta_{0}=30{}^{\circ}$, and (**c**) $\theta_{0}=60{}^{\circ}$.

**Figure 7 sensors-22-05456-f007:**
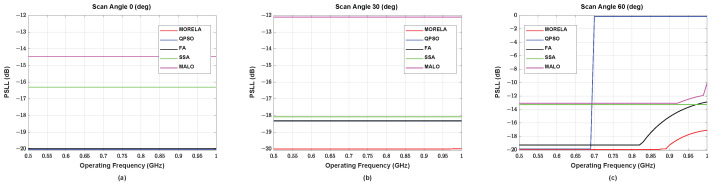
The SLL performance of symmetric NUSLA in the case for (**a**) $\theta_{0}=0{}^{\circ}$, (**b**) $\theta_{0}=30{}^{\circ}$, and (**c**) $\theta_{0}=60{}^{\circ}$.

**Table 1 sensors-22-05456-t001:** NUSLA parameter setting.

NUSLA Parameter	Value
Minimum/maximum frequency $\left(f_{min}/f_{max}\right)$	0.5/1 GHz
The number of frequencies $\left(N_{f}\right)$	11
Minimum/maximum spacing $\left(d_{min}/d_{max}\right)$	0.5/1.5 $\lambda_{min}$
Minimum/maximum amplitude $\left(a_{min}/a_{max}\right)$	0.1/1
Minimum/maximum phase $\left(\phi_{min}/\phi_{max}\right)$	0/π
Propagation speed (c)	$3\times 10^{8}$ m/s

**Table 2 sensors-22-05456-t002:** MORELA parameter setting.

MORELA Parameter	Value
The number of maximum learning episode (T)	1000
Size of the sub-environment (K)	20
Learning rate (α)	0.8
Discounting factor (γ)	0.2
Search space reducing factor (β)	0.99

**Table 3 sensors-22-05456-t003:** Performance comparison for symmetric NUSLA.

		Max HPBW (deg)	Min HPBW (deg)	PSLL (dB)	Time (s)
	$\theta_{0}=0{}^{\circ}$	6.16	3.08	−20.15	
MORELA	$\theta_{0}=30{}^{\circ}$	9.74	4.86	−19.99	37.95
	$\theta_{0}=60{}^{\circ}$	22.20	10.52	−18.42	
	$\theta_{0}=0{}^{\circ}$	6.34	3.16	−20.04	
QPSO	$\theta_{0}=30{}^{\circ}$	9.82	4.90	−19.99	20.07 *
	$\theta_{0}=60{}^{\circ}$	16.55	8.05	−0.15	
	$\theta_{0}=0{}^{\circ}$	6.32	3.16	−20.05	
FA	$\theta_{0}=30{}^{\circ}$	10.43	5.21	−19.98	194.98
	$\theta_{0}=60{}^{\circ}$	22.97	10.84	−17.90	
	$\theta_{0}=0{}^{\circ}$	6.48	3.24	−20.00	
SSA	$\theta_{0}=30{}^{\circ}$	11.50	5.74	−14.67	18.87
	$\theta_{0}=60{}^{\circ}$	21.18	10.10	−13.47	
	$\theta_{0}=0{}^{\circ}$	10.38	5.18	−18.63	
MALO	$\theta_{0}=30{}^{\circ}$	12.32	6.14	−15.18	47.50
	$\theta_{0}=60{}^{\circ}$	22.97	10.84	−14.19	

* The presented runtime of QPSO is with the optimal hyper-parameter setting found by several trial-and-error. QPSO significantly depends on the initial hyper-parameter and at least 4~10 additional trials of QPSO is needed
to find an optimal setting.

**Table 4 sensors-22-05456-t004:** Performance comparison for asymmetric NUSLA.

		Max HPBW (deg)	Min HPBW (deg)	PSLL (dB)	Time (s)
	$\theta_{0}=0{}^{\circ}$	6.00	3.00	−20.04	
MORELA	$\theta_{0}=30{}^{\circ}$	9.91	4.96	−20.01	117.56
	$\theta_{0}=60{}^{\circ}$	21.78	10.46	−17.08	
	$\theta_{0}=0{}^{\circ}$	6.22	3.10	−20.03	
QPSO	$\theta_{0}=30{}^{\circ}$	10.11	5.04	−18.34	53.40
	$\theta_{0}=60{}^{\circ}$	14.42	7.13	−0.15	
	$\theta_{0}=0{}^{\circ}$	6.54	3.28	−19.97	
FA	$\theta_{0}=30{}^{\circ}$	10.84	5.42	−18.32	520.75
	$\theta_{0}=60{}^{\circ}$	22.18	10.55	−12.87	
	$\theta_{0}=0{}^{\circ}$	6.58	3.28	−16.30	
SSA	$\theta_{0}=30{}^{\circ}$	11.03	5.51	−18.08	52.44
	$\theta_{0}=60{}^{\circ}$	21.10	10.07	−13.26	
	$\theta_{0}=0{}^{\circ}$	9.96	4.98	−14.47	
MALO	$\theta_{0}=30{}^{\circ}$	11.14	5.55	−12.11	136.30
	$\theta_{0}=60{}^{\circ}$	21.06	10.1	−10.06	

## Data Availability

This study did not report any data.

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
