# Peer review of "Design Method for a Wideband Non-Uniformly Spaced Linear Array Using the Modified Reinforcement Learning Algorithm"

_sensors, 2022, doi:10.3390/s22145456_

Round 1

Reviewer 1 Report

In this research article the author has presented Design Method for a Wideband Non-Uniformly Spaced Linear Array Using the Modified Reinforcement Learning Algorithm. The topic is interesting however authors need to modify some issues like

1.       Author should clearly mention the novelty of this article.

2.       The author should added some more recent literature and compare his results with existing published article Based on various parameters.

3.       5.2. Wideband asymmetric NUSLA  more explanation need to add

4.        Why the author used normalized radiation pattern?

5.       The author need to add a schematic layout of the antenna/

6.       What are the benefits of using this method through MATLAB in lieu of CST and HFSS simulation software?

7.       Why the author called the experiments results? Why not only simulation?

Author Response

Authors deepley appreicate valuable comments from the reviewer.

Reviewer 2 Report

This paper presents an optimization algorithm based on reinforcement learning (RL) that allows optimizing both spacing and weights. As a novelty, the authors claim to propose a cost function that penalizes only the sidelobe level and not the beam pattern, which increases degrees of freedom of weights and, therefore, the designing process. The proposed optimization algorithm could have a potential interest from researchers. However, there are questions about the presented work that the authors could hopefully address:

  • The main idea of Fig. 2 is not clear to me. What are the red areas? Besides the beam pattern, I do not see the differences.
  • The comparison procedure lacks details. Did authors use their implementations of each method and run it on the same system?
  • By looking at the comparison table, I do not see a game-changing performance. The authors claim that they reduce the complexity of the optimization, but without giving details, I cannot find the proof in the text.
  • The latter argument puts the significance of the whole paper in question. Is it a significant improvement to be interesting for other researchers? I suggest that the authors address this point in the main text before publishing. Or kindly explain what this reviewer missed. 

Author Response

(The authors gave the same response as above.)

Round 2

Reviewer 2 Report

I appreciate the effort made by the authors to improve the work. However, I insist that the conclusions do not fully reflect the results obtained in this paper. As it is right now, the conclusion part is scarce. What is the problem if the complicated hyper-parameter setting is still done faster than RL? If the results of QPSO highly depend on the initial settings, then the authors should mention this. Overwise, the slight gain in HPBW and PSLL does not justify running longer time.

As a minor comment, the new figure of the antenna array needs some explanation (some legends, probably).

Author Response

Authors deeply appreicate valuable comments from the reviewer.

Round 3

Reviewer 2 Report

I would recommend authors eliminate a slight contradiction in their statements regarding the performance of QPSO. For instance, they can put a footnote in Table 4 for QPSO, stating that this time can vary significantly depending on the initial hyperparameters. Also, if possible, mention the magnitude of such a variation. Are we talking about two or ten trials needed to reach this time?

Author Response

All authors deepley appreicate valuable comments from the reviewer.
